# Spatial Distribution Characteristics of Public Fitness Venues: An Urban Accessibility Perspective

Yong Jiang [1], Yangyang Liu [1], Zelei Liu [1], Chunwei Wang [1], Zhipeng Shi [1], Hongbo Zhao [1,*], Dongqi Sun [2], Wei Sun [3,*] and Xiangquan Wang [4]

1. School of Physical Education, Liaoning Normal University, No. 850 Huanghe Road, ShaHekou District, Dalian 116029, China
2. Key Laboratory of Regional Sustainable Development Modeling, Institute of Geographic Sciences and Natural Resources Research, CAS, Beijing 100101, China
3. Nanjing Institute of Geography and Limnology, Key Laboratory of Watershed Geographic Sciences, Chinese Academy of Sciences, Nanjing 210008, China
4. College of Physical Education, Jilin University, Changchun 130012, China
* Correspondence: hongbo1128@lnnu.edu.cn (H.Z.); wsun@niglas.ac.cn (W.S.)

**Abstract:** In the context of healthy China, the study of the spatial distribution characteristics of urban sports venues is not only beneficial to planning the construction of sports venues in cities, but also to the health protection of urban residents. Therefore, to promote a fair and scientific approach to constructing public fitness sites in the city and meet the needs of urban residents' fitness activities, this study targeted public fitness sites larger than 10 m$^2$ in the main urban area of Jinan City to study spatial distribution characteristics and accessibility. We combine the traffic road network and other data and use spatial and buffer zone analyses to assess the sites from the perspective of different travel modes of urban residents. The results show that the public fitness venues in the main urban area are mainly concentrated centrally; there is no significant pattern between the construction of venue area and population. For the time range of 0–15 min, Lixia District has the highest ratio of public fitness venue service area for walking, cycling, and car travel, with 22.54%, 62.25%, and 100%, respectively, and Changqing District has the lowest. In terms of travel mode, the highest service area ratio is 62.7% for car travel, followed by 28.7% for cycling, and 7.7% for walking. It is concluded that the construction of public fitness venues in Jinan has an unbalanced layout, does not fully consider the population factor, and different modes of travel have a significant impact on accessibility. It is therefore suggested that the government should increase public fitness venue construction in the areas surrounding the main city; moreover, future planning of urban public fitness venues should fully consider the distribution characteristics of population quantity and age in each area. Finally, the main travel mode characteristics of urban residents should also be considered to promote the future scientific development of urban public fitness venue construction.

**Keywords:** public fitness venues; spatial distribution characteristics; accessibility; Jinan city

## 1. Introduction

Public fitness venues are public facilities for fitness where urban residents conduct sports activities. These venues play an important role in the national fitness activities carried out in cities [1]. Certain studies have shown that active participation in physical activity is beneficial to the promotion of physical health [2]. Inadequate physical activity plays a key role in the development of non-communicable diseases and can adversely affect mental health and quality of life [3,4]. The rapid development of the urban economy and the residents' increased pursuit of health [5] have led to increased demand for physical activity and the corresponding facilities [6]. As public fitness venues are the main facilities for physical activities among urban residents, they are a significant influencing factor for the level of physical activity and health among urban residents. Moreover, certain studies

have found that public fitness venues in cities exhibit uneven spatial distribution and unbalanced supply [7]. Therefore, based on the Healthy China strategy, the study of the spatial distribution characteristics of public fitness venues in cities, which is beneficial to the benign development of urban physical activity spaces and the health promotion of urban residents, is important [8].

Scholars of urban public fitness venues have studied the linkages between urban public fitness venues and urban public spaces [9], COVID-19 [10], public health, urban green spaces [11] and equity. For example, Liu et al. [12] assessed the equity of urban residents' access to public fitness venues based on the link between urban public sports facilities and public health, and found that urban centers have more space resources for physical activity than urban fringe areas. Yi and Horton [13] proposed that China has abundant open green spaces and residential parks whose potential value for fitness and sports has not been fully explored, and that effective integration of the quantity and quality of garden green belts, residential green spaces, and sports fields can promote the future level of utilization of green space resources in China. In addition, other scholars have studied public fitness venues from the perspectives of their social attributes [14], spatial distribution [15,16], and accessibility [17]. By studying the spatial distribution of gymnasiums in Germany, Wu et al. [18] found that the main factor affecting residents' satisfaction with fitness venues is accessibility, followed by conditions such as fresh air and clean environment. Therefore, accessibility to sports venues in Germany is limited [19]. From the perspective of urban traffic accessibility, based on the road network and considering the different travel modes of urban residents, this study investigated the service area of public fitness venues within 15 min, to explore the construction of accessibility of urban public fitness venues.

Accessibility refers to the ease of communication between different spatial entities thus overcoming distance barriers [20], and its main calculation methods are the buffer zone analysis, cost distance, and gravitational model methods. The buffer zone analysis method is essentially based on the topological data of the map patch type and calculates the range covered by polygons that are situated at a certain distance from a certain geographical target. The cost-distance method mainly calculates the time or material consumed (generally time and money) from a particular point or region to a target point, and abstracts the targets as points in the evaluation process, which leads to a relatively large error in the evaluation results; therefore, the buffer zone analysis method was selected for this study. The gravitational model method is suitable for urban scale or regional spatial scale studies which do not require high accuracy of results. In contrast, the buffer zone analysis method can consider the influence of road networks, thus its results more clearly express the accessibility of a certain service facility.

Accessibility analysis has been widely applied geographically, such as for transportation planning [21] and urban building planning [22] in cities. Various scholars have studied the relationship between accessibility to urban transportation and urbanization level [23], urban bus speed, level of road infrastructure, location of public service facilities, and level of administrative district. Zuo et al. [24] found that the adjustment of bus schedules according to time-varying travel demand affects the accessibility of the transit system, and that land use development, average bus speed, and transit facilities all positively affect the accessibility of the transit system. Petras and Kveton [25] found that the impact of new road infrastructure on accessibility in the Czech Republic is strong in a few areas, whereas the impact of new road infrastructure on traffic load is relatively evenly distributed spatially. In addition, certain scholars have conducted accessibility analyses of population, hydrology, urban parks, and green spaces in cities and determined the optimal travel time for urban residents and the spatial distribution characteristics of urban green and blue spaces [26,27]. Therefore, accessibility analysis can be used to objectively evaluate the fairness and convenience of the spatial distribution of urban parks, green spaces, transportation, and residential buildings. Based on the above results and conclusions of accessibility research, this study applies the concept of accessibility research to the field of urban public fitness venue construction and investigates the accessibility of public fitness venues based

on road network, so as to analyze the differences in accessibility of public fitness venues in different regions.

Research perspectives on the accessibility of sports activity venues are diverse and include the analysis of the spatial accessibility of sports facilities, and research on the relationship between accessibility of sports facilities and physical activity, and economic factors. For example, Lee et al. [28] found that accessibility of sports facilities was significantly associated with physical activity among Korean adults. In terms of the relationship between accessibility of sports activity venues and regions, Xiao et al. found that the accessibility of sports facilities in northern Dongguan was higher than that in southern Dongguan based on the MG2SFCA method [29]. Higgs et al. [30] performed accessibility analysis and found that people in poorer areas of Wales had more opportunities to access public sports facilities.

In a study on accessibility based on different modes of travel, Pinto found significant differences between the accessibility of green spaces in two European cities when urban residents chose different modes of transportation. Car travel was found to be highest in both cities [31]. The use of private cars in Oforikrom District is more optimized than that of public transport in terms of accessibility and travel time according to a related study. Liang used a multiple regression model to calculate the average weighted walking time from road intersections and neighborhood entrances to urban park entrances to assess walkability in the urban parks in Shanghai. Park service areas, walkability, and supply efficiency varied significantly according to subdistrict [32]. Therefore, the mode of travel is an important factor affecting the accessibility of public fitness venues. One study showed that in 2015, in Jinan city alone, 43.3% of urban residents chose transportation by private car, 19.5% by bus, and 28.9% by walking and cycling.

Therefore, the buffer zone analysis method was used in this study to calculate the service area of public fitness venues in each district within 0 to 15 min and the service area ratio based on the residents' walking, cycling, and car travel, to analyze the accessibility of the 15 min fitness circle in each urban area.

## 2. Materials and Methods

### 2.1. Study Area

Jinan City is located in the central part of Shandong Province, between 36°02′ and 37°54′ N latitude and 116°21′ and 117°93′ E longitude. Jinan City has high topography in the south and low topography in the north and is within the warm temperate continental monsoon climate zone with four distinct seasons. As shown in Figure 1, Huaiyin, Tianqiao, Shizhong, Changqing, Lixia, and Licheng districts of Jinan City were selected as the study area.

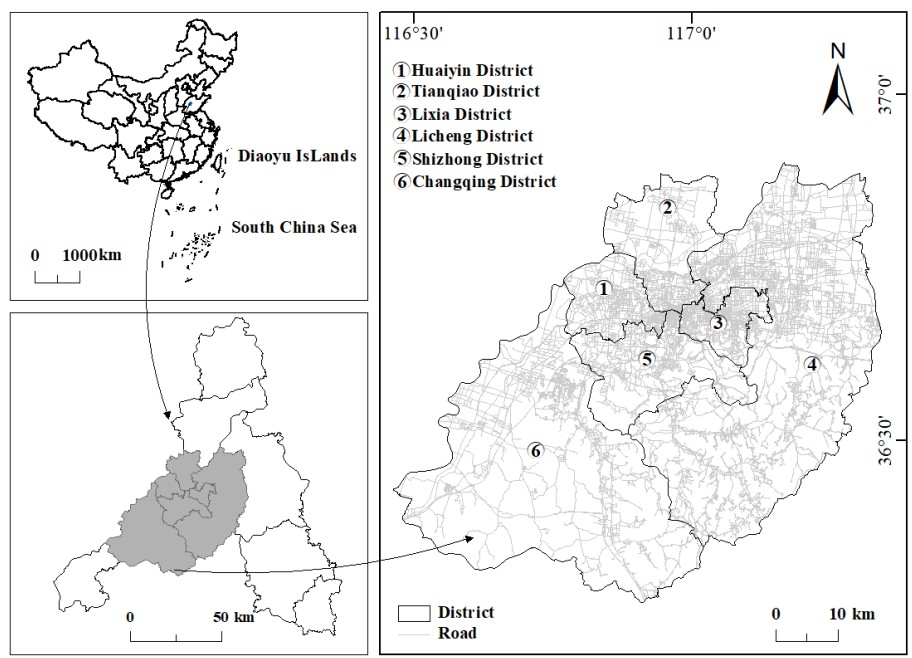

**Figure 1.** Location of the study area.

### 2.2. Headings

The names and areas of public fitness venues used in this study were obtained from the Jinan Sports Bureau, and data on the latitude and longitude of public fitness venues were obtained from Baidu Maps. In this study, the identified public fitness sites were screened according to their level, area, and visualization in the map. Ultimately, 2900 public fitness sites covering an area of $\geq 10$ m$^2$ were selected as the study sites. The administrative vector data on the main urban area of Jinan City were obtained from the Resource and Environment Science and Data Center, as well as the Baidu Maps website. The scope and boundary of each administrative district, and the road network data for Jinan City were obtained from Openstreetmap (OSM), as shown in Table 1.

**Table 1.** Data sources and methods.

| Data Name | Data Description | Data Sources |
|---|---|---|
| Data of public fitness venues | name, area and location | Jinan Sports Burea (http://jnstyj.jinan.gov.cn/,accessed on 11 August 2022), Baidu map (https://www.baidu.com/, accessed on 20 August 2022) |
| Jinan Road network data | Road name, road distribution | Openstreet map (https://www.openstreetmap.org, accessed on 13 September 2022) |
| Data of Jinan main urban area | Main urban boundary | Resource and Environmental Sciences and Data Center (https://www.resdc.cn/, accessed on 11 September 2022) |

Based on the railroad mileage and speed standards of different levels in China, the Technical Standards for Highway Engineering of the People's Republic of China (JTG B01-2014), and previous study results, the average walking speed in Jinan urban road network in this study was determined to be 5 km/h, and the average cycling speed 15 km/h [33]; the average car travel speed is shown in Table 2.

**Table 2.** Average speed of traffic on main city roads in Jinan.

| Road | Speed (km/h) | Road | Speed (km/h) |
|---|---|---|---|
| Motorway | 100 | Primary_link | 40 |
| Motorway_link | 40 | Secondary | 50 |
| Trunk | 70 | Secondary_link | 40 |
| Trunk_link | 40 | Tertiary | 45 |
| Primary | 55 | Tertiary_link | 40 |

*2.3. Methods*

This study is based on data of the main urban area boundary, urban road network, and the location and area of public fitness venues in Jinan City. Spatial overlay, buffer zone, and comparative analyses were used to analyze the spatial distribution characteristics of public fitness venues in the main urban area of Jinan City. In addition, based on the road network data on the main urban area of Jinan City and using the buffer zone calculation method, the service areas of public fitness venues under each travel mode within 0–5 min, 0–10 min, and 0–15 min were calculated based on three modes of transport: walking, cycling, and car travel, respectively. Then, their service areas were determined and their service area ratios were calculated using the following formula:

$$P = S_1/(S_2 - S_3) \times 100\% \tag{1}$$

where P represents the ratio of public fitness venue service area; $S_2$ is the area of each administrative district; $S_3$ represents the area of public fitness venues in each administrative district; and $S_1$ is the area of the public fitness venue service area. The formula calculates the ratio of public fitness venue service area for each administrative district. The work flow chart is shown in Figure 2.

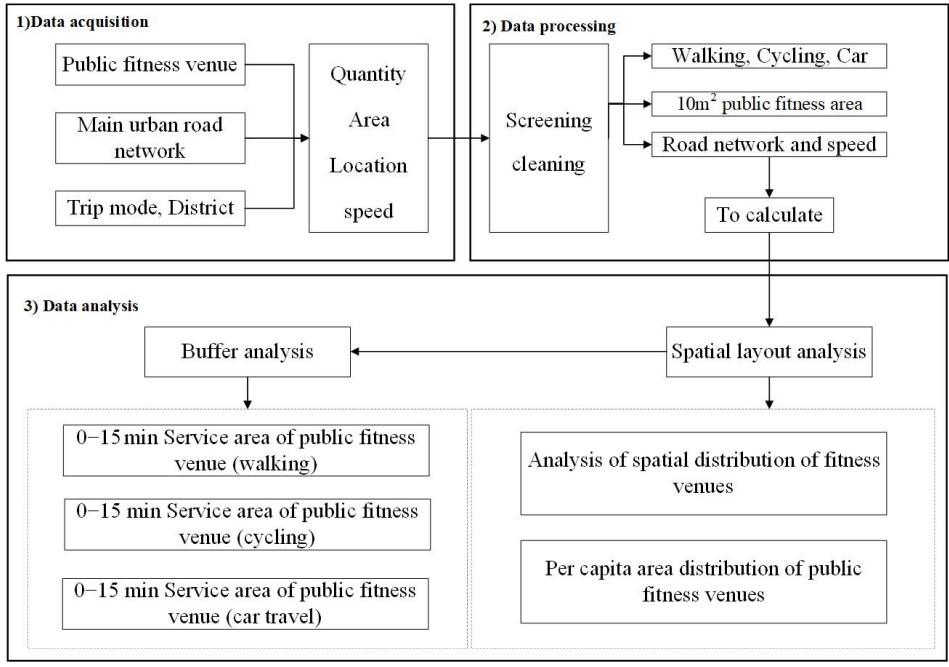

**Figure 2.** Work flow chart.

## 3. Results

*3.1. Public Fitness Venue Location Distribution Characteristics*

To explore the spatial distribution characteristics of public fitness venues in the main urban area of Jinan, ArcGIS10.7 software was used to visualize the coordinate location distribution of public fitness venues in the main urban area.

As seen in Figure 3, public fitness venues covering than 10 m² are mainly concentrated in the central area of the main city of Jinan, and the number of public fitness venues gradually decreases from the center to the surrounding area. As shown by Table 3, Licheng District has the largest number of public fitness venues (612). However, owing to the large area of the city, the southern and central parts of the city are mostly forested and mountainous; therefore, the public fitness venues exhibit a scattered distribution and are mainly concentrated in the northern and southeastern parts of the city. The number of public fitness venues in the city's central district is second to that of the city's calendar district, a total of 610, and clear aggregation is observed. The public fitness venues are mainly concentrated in the northern part of the city and scattered in the other parts. The number of public fitness venues in Changqing District is 585, and because the terrain is mostly covered by mountains and dense forest, the distribution is mainly concentrated in the southern area, and exhibits obvious aggregation. In Tianqiao District, there are 563 public fitness venues. The northern part of Tianqiao District is mostly covered by cultivated land, and the southern part by mature urban development with a high concentration of residents. Therefore, public fitness venues are mainly concentrated in the south, particularly the areas bordering Central City and Lixia Districts. Therefore, Tianqiao District exhibits uneven distribution of public fitness venues from north to south. Huaiyin District has 308 public fitness venues, and they are relatively scattered, with no aggregation characteristics. Lastly, there are 222 public fitness venues in the northern and western regions of Lixia District, the least in the district; particularly in the areas bordering the city and Tianqiao District. The number of public fitness venues in the northeastern and southeastern regions is low. The reason for this is that the mountainous and hilly terrain in the southwest and southeast of the main urban area of Jinan is prominent, and the comparison between the urbanization level and central area is generally low, so the spatial distribution of public fitness venues around the main urban area is scattered as a whole and clustered in a specific area.

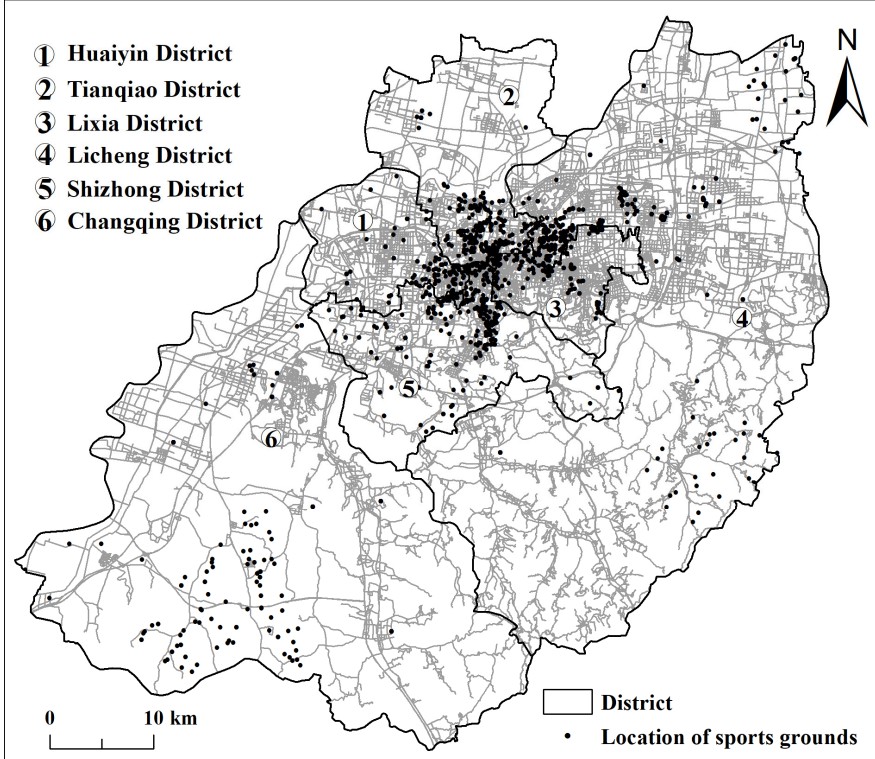

**Figure 3.** Distribution of public fitness venues in the main urban area of Jinan.

**Table 3.** Number of public fitness venues in the main urban area of Jinan.

| District | Tianqiao | Lixia | Shizhong | Changqing | Licheng | Huaiyin |
|---|---|---|---|---|---|---|
| Quantity | 563 | 222 | 610 | 585 | 612 | 308 |
| Sports area (km$^2$) | 0.08 | 0.03 | 0.09 | 0.08 | 0.09 | 0.15 |

*3.2. Distribution Characteristics of per Capita Area of Public Fitness Venue*

Per capita public fitness venue is an important reference to measure the construction of public sports and fitness venues in a certain area; therefore, it is necessary to analyze the relationship between the permanent population of the main urban area of Jinan City and the area of public fitness venues. This study calculates the per capita public fitness venue area of each district, and compares and analyzes the construction of public fitness venues in the main urban area of Jinan City from the perspective of the population factor. According to Table 4, the largest resident population in the main urban area of Jinan is Licheng District, with a total of 1,112,022 residents, followed by Lixia District, Shizhong, Changqing, and Tianqiao districts, and Huaiyin District has the smallest permanent population. However, the largest area of public fitness venues is Huaiyin District, with 150,666 m$^2$, while Lixia District has the lowest, only 26,919 m$^2$, which shows that the construction of public fitness venues in the main urban area of Jinan City does not match the distribution characteristics of the resident population of each district. By calculating the per capita area of public fitness venues in each district, it was found that Huaiyin District has the highest per capita fitness venue area of 0.22 m$^2$, followed by Changqing, Tianqiao, and Shizhong districts, all exceeding 1, 0.13, 0.12, and 0.10 m$^2$, respectively, and finally Licheng and Lixia districts are both below 1 m$^2$, only 0.08 and 0.03 m$^2$, respectively. Based on this, we find that there is no certain law or connection between the area construction of public fitness venues in the main urban area of Jinan City and the population factor. For example, Lixia District has the lowest per capita public fitness venue, and the construction of public fitness venues in this area should be increased. Therefore, population should be an important factor considered in the planning and construction of public fitness venues to meet the needs of urban residents' fitness activities. Thus, it is recommended that the government focus on the actual distribution of the region's population and age characteristics when building and planning public fitness venues in the future.

**Table 4.** Statistics of the area of public fitness places per capita in the main urban area of Jinan City.

| District | Huaiyin | Tianqiao | Shizhong | Changqing | Lixia | Licheng |
|---|---|---|---|---|---|---|
| The resident population | 675,048 | 718,024 | 903,714 | 595,549 | 819,139 | 1,112,022 |
| Public fitness area (m$^2$) | 150,666 | 84,433 | 94,698 | 77,077 | 26,919 | 89,498 |
| Per capita site area (m$^2$) | 0.22 | 0.12 | 0.10 | 0.13 | 0.03 | 0.08 |

*3.3. Accessibility Analysis of Public Fitness Venues*

In this study, buffer zone analysis was performed on the accessibility of public fitness venues. The service area of public fitness venues refers to the coverage area of each public fitness venue as the center, according to the three modes of transport among urban residents: walking, cycling, and car travel within 0–15 min of travel [34]. The analysis is discussed below.

3.3.1. Transportation by Walking

As shown in Figure 4 and Table 5, within the time range of 0–5 min based on walking as a mode of transport, the service area and service area ratio of public fitness venues in Lixia District are 12.34 km$^2$ and 12.18%, respectively, thus ranking highest. This is followed by Shizhong, Zhongshan, Tianqiao, Huaiyin, Licheng, and Changqing districts. The service area of public fitness venues in Changqing District is 10.19 km$^2$. However, its urban area covers up to 1217.54 km$^2$, thus it has the lowest service area ratio for public fitness venues,

0.84%. Within the time range of 0–10 min, the service area ratio of fitness venues in each district was observed to improve. The service area of Lixia District is 22.82 km$^2$, and the service area ratio increased to 22.54%, hence it maintained the highest ranking. This was followed by the city of Central District with a service area ratio of 14.9%, and then Huaiyin and Tianqiao districts, with service area ratios of 13.45% and 12.79%, respectively. Licheng and Changqing districts ranked last, with service area ratios of 0.84% and 1.39%, respectively. Within the 0–15 min time range, the service area ratio of public fitness venues in Lixia District rose to 32.45%, thus maintaining the highest ranking. Those of central City and Huaiyin Districts rose from 14.19% and 13.45%, respectively, to 20.16%, thus jointly ranking second. Those of Tianqiao, Licheng, and Changqing districts followed in ranking, with service area ratios of 17.19%, 5.01%, and 1.88%, respectively. Although the service area of public fitness venues in Changqing District was 22.93 km$^2$, the service area ratio of its fitness venues remained relatively low.

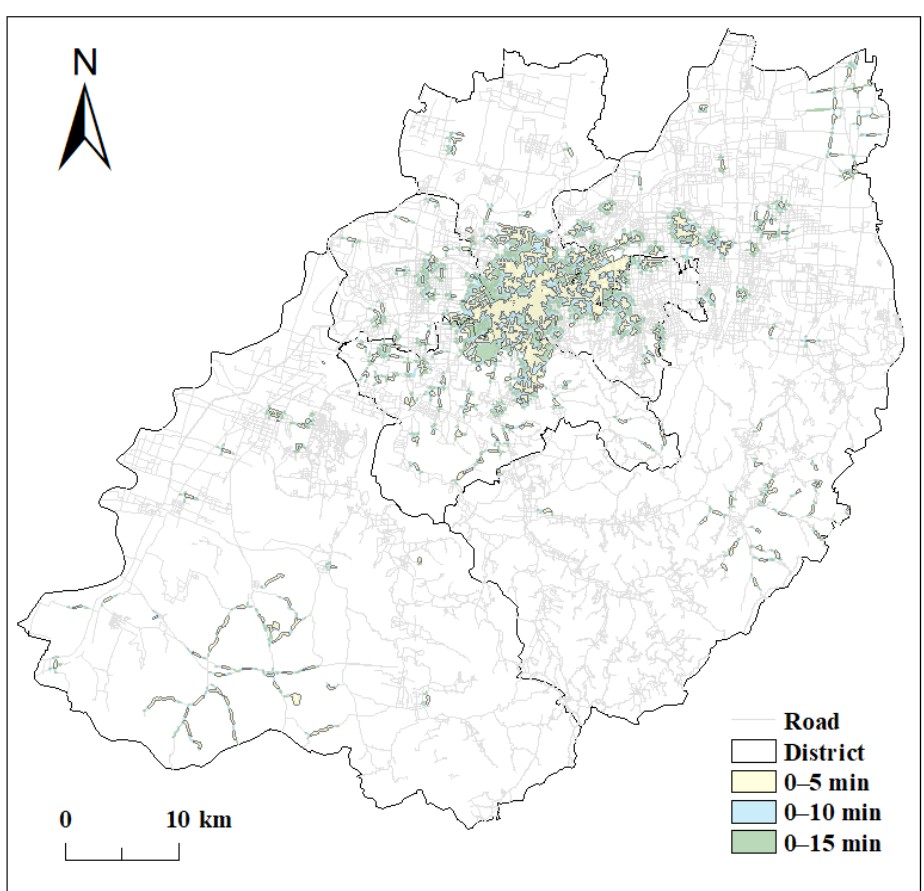

**Figure 4.** Service area scope of public fitness venues in the main urban area of Jinan with respect to walking as transportation.

**Table 5.** Service area and service area ratio of public fitness venues in the main urban area of Jinan City (walking).

| The City Name | Urban Area/(km$^2$) | Site Area (km$^2$) | The Service Area/(km$^2$) | | | Service Area Ratio/(%) | | |
|---|---|---|---|---|---|---|---|---|
| | | | 0–5 | 0–10 | 0–15 | 0–5 | 0–10 | 0–15 |
| Tianqiao District | 261.58 | 0.08 | 19.88 | 33.44 | 44.95 | 7.60 | 12.79 | 17.19 |
| Lixia District | 101.29 | 0.03 | 12.34 | 22.82 | 32.86 | 12.18 | 22.54 | 32.45 |
| Shizhong District | 290.01 | 0.09 | 24.03 | 41.15 | 58.45 | 8.29 | 14.19 | 20.16 |
| Changqing District | 1217.54 | 0.08 | 10.19 | 16.92 | 22.93 | 0.84 | 1.39 | 1.88 |
| Licheng District | 1312.41 | 0.09 | 24.66 | 44.23 | 65.77 | 1.88 | 3.37 | 5.01 |
| Huaiyin District | 151.72 | 0.15 | 11.35 | 20.38 | 30.55 | 7.49 | 13.45 | 20.16 |

Based on these results, the highest service area ratio of public fitness venues within 0–15 min walking time for residents in the main urban area of Jinan is observed in Lixia District, followed by Shizhong, Huaiyin, Tianqiao, Licheng, and Changqing districts. Table 5 shows that despite the significant differences among the service areas and service area ratios of fitness venues of the main urban areas of Jinan, the overall service area ratio is low, the highest being only 32.45%. Therefore, the accessibility of public fitness venues within 0–15 min for residents on foot in the main urban areas of Jinan is low and fairly distant from the expected full coverage of the 15 min fitness circle.

The emergence of this phenomenon is related to the unique advantages of the urban area, for example: Lixia District, as the old town of Jinan, has a large number of tourist attractions and universities, a relatively complete planning of pedestrian roads, and a high density of public fitness venues, resulting in the largest range of public fitness services in Lixia District. In contrast, Changqing District is subject to certain restrictions on traffic road planning due to its prominent mountainous and hilly landforms, and the public fitness venue facilities are highly dispersed and not at walking distances. Therefore, the proportion of public fitness service areas in Changqing District is the lowest.

### 3.3.2. Transportation by Cycling

As shown in Figure 5 and Table 6, within the time range of 0–5 min based on the cycling mode of transport, the service area of public fitness venues in Lixia District was 42 km², with a service area ratio of 41.48%, thus ranking highest. This was followed by Huaiyin and Shizhong districts, whose service area ratios are 27.34% and 25.42%, respectively. Tianqiao District ranked after Zhongcheng District, with a service area of 261.58 km² for public fitness venues and a service area ratio of 19.29%. Licheng and Changqing districts, whose public fitness venues have a service area exceeding 1000 km², and whose service area ratios were 7.07% and 2.46%, respectively, ranked lowest. Within the 0–10 min time range, the service area ratio of fitness venues in each district was observed to improve. Lixia and Huaiyin districts ranked first and second, with their service area ratios exceeding 50%: 62.25% and 54.66%, respectively. They were followed in ranking by Central District with a service area ratio of 48.90%, exhibiting a slight difference from that of Huaiyin District. In addition, the service area ratio of public fitness venues in Tianqiao and Licheng districts increased to 25.95% and 16.25%, whereas the service area ratio of Changqing District did not exceed 10%; it was 7.54%, thus ranking lowest. Within the 0–15 min time range, the service area ratio public fitness venues in Lixia District remained the highest, reaching 77.5%, whereas those in Huaiyin and Shizhong districts reached 74.38% and 63.20%, respectively. They were followed by Tianqiao, Licheng, and Changqing districts, whose public fitness venue service areas increased to 81.03, 312.2, and 188 km², respectively. Their service area ratios also increased to 30.99%, 23.79%, and 15.44%, respectively.

**Table 6.** Service area and service area ratio of public fitness venues in the main urban area of Jinan City (cycling).

| The City Name | Urban Area (km²) | Site Area (km²) | The Service Area/(km²) | | | Service Area Ratio/(%) | | |
|---|---|---|---|---|---|---|---|---|
| | | | 0–5 | 0–10 | 0–15 | 0–5 | 0–10 | 0–15 |
| Tianqiao District | 261.58 | 0.08 | 50.45 | 67.86 | 81.03 | 19.29 | 25.95 | 30.99 |
| Lixia District | 101.29 | 0.03 | 42.00 | 63.04 | 78.48 | 41.48 | 62.25 | 77.50 |
| Shizhong District | 290.01 | 0.09 | 73.69 | 141.76 | 183.22 | 25.42 | 48.90 | 63.20 |
| Changqing District | 1217.54 | 0.08 | 29.89 | 91.76 | 188 | 2.46 | 7.54 | 15.44 |
| Licheng District | 1312.41 | 0.09 | 92.82 | 213.2 | 312.2 | 7.07 | 16.25 | 23.79 |
| Huaiyin District | 151.72 | 0.15 | 41.44 | 82.84 | 112.73 | 27.34 | 54.66 | 74.38 |

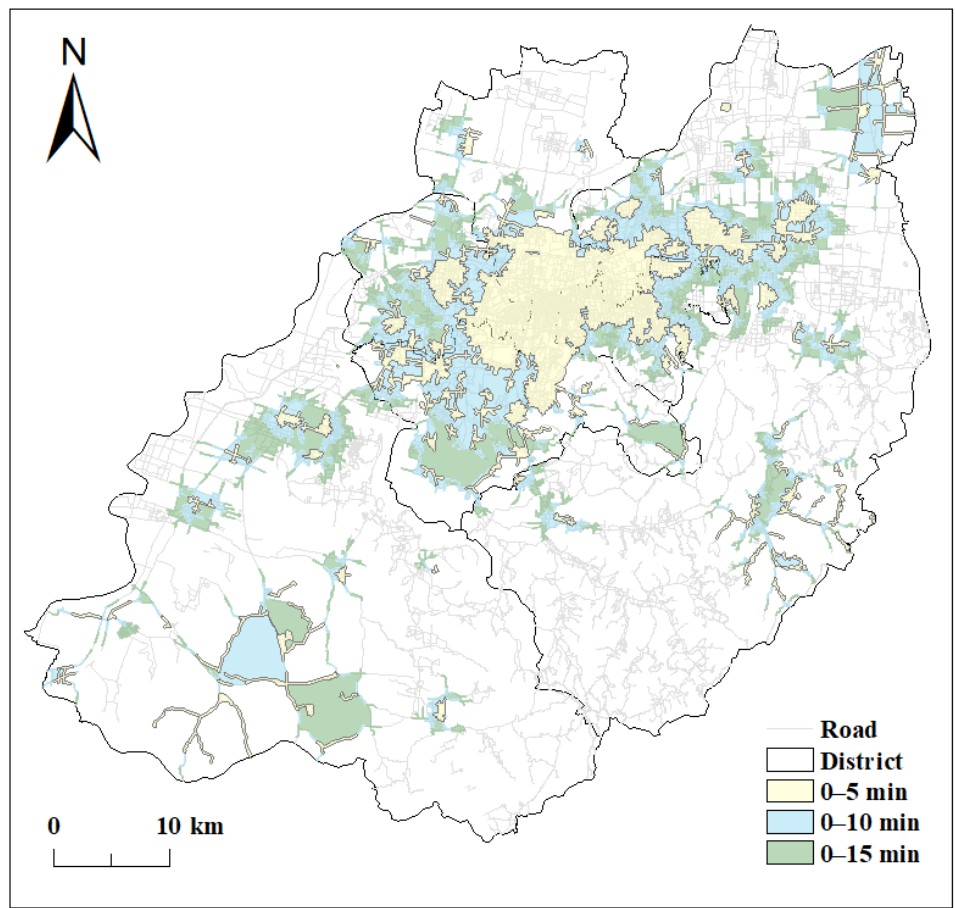

**Figure 5.** Service area scope of public fitness venues in the main urban area of Jinan with respect to cycling as transportation.

The service area ratio of public fitness venues within 0–15 min for residents who chose cycling as the mode of transport in the main urban area of Jinan remained highest in Lixia District, followed by Huaiyin, Shizhong, Tianqiao, Licheng, and Changqing districts. The service area ratios of public fitness spaces in both Lixia and Huaiyin districts exceed 70%, exhibiting significant improvement compared to those of walking.

The reason for this is due to practical factors such as urban congestion; it has become a habit for urban residents to choose cycling. For walking distance comparison, due to the dexterity and convenience of cycling, people can reach destinations faster riding a cycle than walking. In addition, the perfect bicycle road network in the urban area provides support for cycling which is loved by the urban residents, especially in Huaiyin, Lixia, and Shizhong districts, owing to the small area of these districts, the improved bicycle roads, and the dense distribution of public fitness venues. This results in a much higher increase in its service area than other areas. In general, the service range of public fitness venues under cycling has been greatly improved.

### 3.3.3. Transportation by Car Travel

As shown in Figure 6 and Table 7, within the time range of 0–5 min based on car travel, the service area and service area ratios of public fitness venues in Huaiyin District ranked highest at 119.05 km² and 78.55%, respectively. This is followed in ranking by Lixia District with a service area ratio of 76.63%, and then Central City, Tianqiao, and Licheng districts with service area ratios of 56.20%, 27.74%, and 22.10%, respectively. Changqing District at only 10.07% ranks lowest in service area ratio. Within the 0–10 min time range, the service area ratio of the public fitness venues in Lixia District reached 100%, the only administrative district to reach 100%. This indicates that all residents of Lixia District have access to a

public fitness venue within 0–10 min when they choose to travel by car. This was followed by Huaiyin and Shizhong districts, whose service area ratios reached 90.27% and 85.75%, respectively. Lastly, Licheng, Tianqiao, and Changqing districts, whose service area ratios for public fitness venues were 46.74%, 42.92%, and 32.70%, respectively, ranked lowest. However, Changqing District's service area ratio increased significantly. Within the time range of 0–15 min, the service area ratio of public fitness venues in Lixia District remained at 100%. Those of Huaiyin and Shizhong districts are 92.11% and 90.57%, respectively. The service area ratio of Licheng District ranked next at 63.01%, followed by that of Tianqiao District at 56.6%. In addition, the service area ratio of Changqing District exceeded 50% for the first time, rising to 50.24%, but still ranked last. Lixia District has the highest service area ratio of public fitness venues in the main urban area of Jinan: 100%, followed by Huaiyin and Shizhong districts, at >90%, and finally Licheng, Tianqiao, and Changqing districts. Car travel facilitates long-distance travel for city dwellers, and due to the rapid development of the city, there are viaducts and highways in the main urban area to facilitate faster access to their destinations, and due to these road facilities, the time required to reach a destination while driving is faster than walking and cycling. As a result, based on the extensive transportation network, the service area of public fitness venues in Lixia District is 100%. At the same time, because the choice of car travel reduces the consideration of the problems of large discrete distribution and long distance of public fitness venues, the service area ratio of Changqing and Licheng districts increased significantly. However, the above calculations all take into account the impact of factors such as rush hour, driving congestion, and waiting for traffic lights, which is also the direction of our future research.

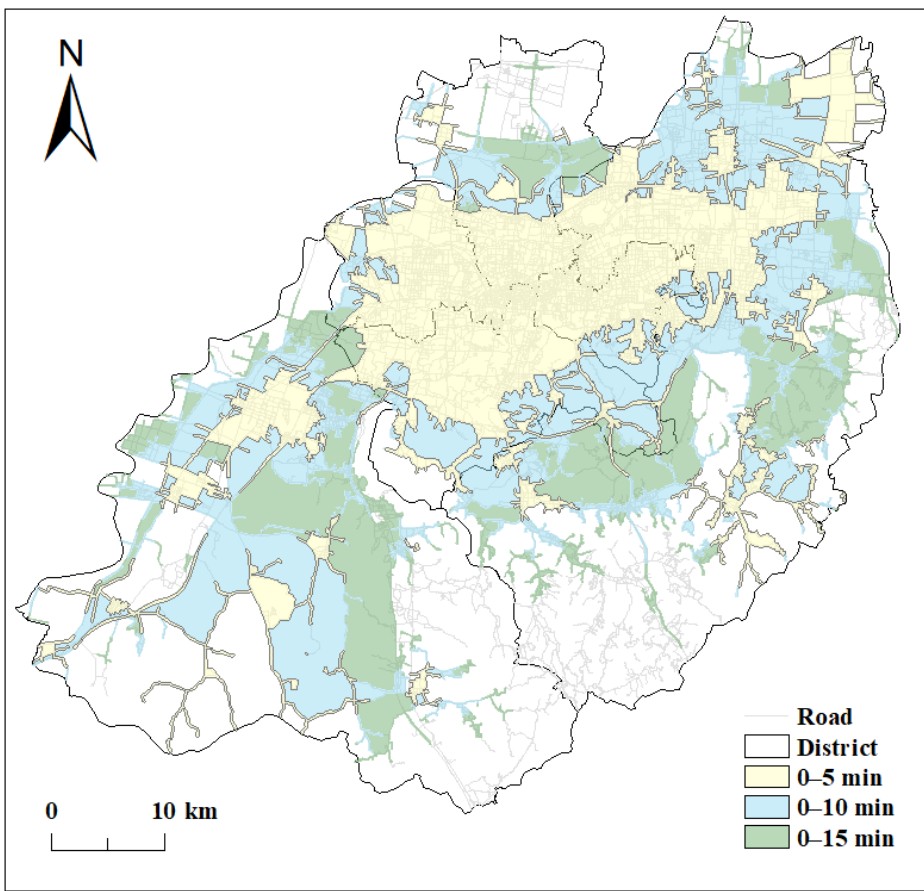

**Figure 6.** Service area scope of public fitness venues in the main urban area of Jinan with respect to car transportation.

**Table 7.** Service area and service area ratio of public fitness venues in the main urban area of Jinan City (car travel).

| The City Name | Urban Area/(km$^2$) | Site Area (km$^2$) | The Service Area/(km$^2$) | | | Service Area Ratio/(%) | | |
|---|---|---|---|---|---|---|---|---|
| | | | 0–5 | 0–10 | 0–15 | 0–5 | 0–10 | 0–15 |
| Tianqiao District | 261.58 | 0.08 | 72.55 | 112.22 | 148.01 | 27.74 | 42.92 | 56.60 |
| Lixia District | 101.29 | 0.03 | 77.60 | 101.29 | 101.29 | 76.63 | 100 | 100 |
| Shizhong District | 290.01 | 0.09 | 162.94 | 248.61 | 262.57 | 56.20 | 85.75 | 90.57 |
| Changqing District | 1217.54 | 0.08 | 122.66 | 398.14 | 611.69 | 10.07 | 32.70 | 50.24 |
| Licheng District | 1312.41 | 0.09 | 289.99 | 613.36 | 826.92 | 22.10 | 46.74 | 63.01 |
| Huaiyin District | 151.72 | 0.15 | 119.05 | 136.82 | 139.6 | 78.55 | 90.27 | 92.11 |

## 4. Discussion

This study was based on three modes of transportation: walking, cycling, and car travel. The distribution of public fitness venues in the main urban area of Jinan was found to be uneven, and the construction and accessibility of the 15 min fitness circles varied. Table 8 shows that the ranking of service areas of the 15 min fitness circle of public fitness venues in the main urban area of Jinan with respect to car travel, cycling, and walking as transportation. The service area ratios with respect to car travel, cycling, and walking are 62.67%, 28.67%, and 7.66%, respectively, and each district exhibits the same characteristics. Evidently, residents' choice of transportation mode can cause significant differences in the accessibility of public fitness venues in the main urban area of Jinan City. This result corresponds to previous research conclusions, as follows. Taking Dongguan as an example, Xiao et al. [29] studied the spatial accessibility of sports facilities for different age groups of urban residents using walking, cycling, and driving as a mode of transport, and found that there were significant spatial differences in the accessibility of sports facilities, with better accessibility in the north and poor accessibility in the south. At the same time, it is found that the accessibility distribution of sports facilities while choosing walking mode of transport shows the greatest spatial difference, while the accessibility in driving mode of travel is the most balanced, which confirms that different travel modes have a significant impact on the accessibility of sports facilities. In the study of the spatial layout of urban sports facilities in Jinan, Yuan et al. [35] proposed that the layout planning of public sports facilities should follow the corresponding models and principles. The layout of public sports facilities at all levels should fully consider factors such as urban public transportation and the reasonable service radius of public sports facilities. However, this study remains in the theoretical stage, and did not conduct microscopic research through data, while the present study is based on the convenience of urban public transportation, considering the diverse choices of urban residents' travel modes. Therefore, the service scope of public fitness venues was studied based on different modes of travel, and the specific data of the service scope of public fitness venues between the main urban areas of Jinan City and the differences between them were obtained. In addition, the present study uses improved methods and considers perspectives on the accessibility of public fitness venues, compared to our previous analysis on the accessibility of public fitness venues in the main urban area of Dalian [36]. However, in that study, only the service scope of public fitness venues under the walking travel mode was calculated, and the impact of different transportation modes of urban residents on the accessibility of public fitness venues was not considered. Therefore, in this study, we combined the actual situation of Jinan residents' travel modes, and we calculated the service range of Jinan public fitness venues for 15 min fitness circles for all three modes of travel: walking, cycling, and car travel. In this way, the difference between the accessibility of public fitness venues under different modes of travel is obtained, which provides a more powerful reference for the construction planning of urban public fitness venues.

**Table 8.** Ratio of public fitness venue service area in the main city of Jinan by different travel modes (%).

|  | 0–5 min | 0–10 min | 0–15 min |
|---|---|---|---|
| Walking trips | 3.07 | 5.36 | 7.66 |
| Cycling trips | 9.91 | 19.80 | 28.67 |
| Car travel | 25.33 | 48.30 | 62.69 |

Based on these findings, this study considers the maximum service area (car travel) of public fitness venues in the main urban area of Jinan City as an example, and analyzes its relationship with the accessibility of public fitness venues in the context of the current situation of population and economic factors in the main urban area of Jinan City, to explore the laws between public fitness venues and population and economic factors, and thus make targeted recommendations. Table 9 shows that, first, from the perspective of population elements, the largest resident population in the main urban area of Jinan is 1,112,022, in Licheng District, but its public fitness service area ratio is 63.01%, and ranks fourth. City Central District, with a resident population of 903,714, ranks second after Licheng District, but its public fitness area service ratio is 90.57%, ranking third. Lixia District has a resident population of 819,139, ranking third, but the ratio of its public fitness venue service area is 100%, the highest administrative district in the main urban area, ranking first. This is followed by Huaiyin, Changqing, and Tianqiao districts with a resident population of 675,048, 595,549, and 718,024, respectively, and their public service area ratios were 92.11%, 50.24%, and 56.60%, ranking second, sixth, and fifth, respectively. Based on this, the Jinan government should plan the construction of public fitness venues in each district based on the distribution of population in each district, to ensure that more fitness venues are available in areas with a relatively large resident population and to avoid the imbalance between the population and the accessibility of fitness venues in each district.

**Table 9.** Statistics on resident population and GDP in the main urban area of Jinan.

|  | Tianqiao | Lixia | Shizhong | Changqing | Licheng | Huaiyin |
|---|---|---|---|---|---|---|
| Population | 718,024 | 819,139 | 903,714 | 595,549 | 1,112,022 | 675,048 |
| GDP (billion) | 64.256 | 191.04 | 116.17 | 37.19 | 116.6 | 70.11 |

From the perspective of economic factors, the highest gross domestic product in the main urban area of Jinan is in Lixha District with 191.04 billion RMB, followed by Licheng, Shizhong, Huaiyin, Tianqiao, and Changqing districts, whereas the ratio of public fitness venue service area from high to low is in Lixha, Huaiyin, Shizhong, Licheng, Tianqiao, and Changqing districts, respectively. The GDP of Licheng District is 116.6 billion RMB and that of Huaiyin District is 70.11 billion RMB, ranking second and fourth, respectively, whereas the ratio of public fitness space service area in Huaiyin District is 92.11% and that of Licheng District is 63.01%, ranking second and fourth, respectively. This shows that there is a mismatch between the GDP of Huaiyin and Licheng districts and the accessibility of public fitness venues. In addition, the GDP of other districts and their accessibility rankings show a pattern of correspondence, for example, the GDP of Central City District ranks third, as does the service area ratio of its public fitness venues. This indicates that the accessibility of public fitness venues is high in areas with high economic development, and the construction of public fitness venues is relatively inferior in areas with low economic development. Evidently, economic factors have a certain correlation with the accessibility of public fitness venues in urban areas, and government departments should reasonably plan the economic expenditure and distribution to increase the construction and planning of public fitness venues.

*Limitations*

Because this study only considers public fitness venues with an area of $\geq 10$ m$^2$ as the original data, the number of public fitness venues used have certain errors compared with the actual situation. Further, this study did not consider the effects of the age or gender of urban residents on walking speed, nor the effects of electric bicycles versus traditional bicycles on riding speed, during the calculation of the service area of public fitness venues. In addition, this study did not consider the influence of urban subways and light railways, nor time spent in road congestion or waiting at traffic lights, on the speed and travel duration of vehicles. Therefore, the service area of public fitness venues calculated in this study likely have some errors compared to the actual situation. In future research, we will consider and address the above limitations to improve the methodological approach.

**5. Conclusions**

Taking the main urban area of Jinan as an example, this study analyzed the service scope and service area ratio of public fitness venues in each district mainly from the perspective of three travel modes using buffer zone, spatial, and comparative analyses. The results show that public fitness venues are mainly concentrated in the central part of the main urban area, and the service area ratio of public fitness venues in the time range of 0–15 min is the highest in Lixia District, while the lowest urban area is Changqing District. In terms of travel mode, the service area ratios of the three travel modes are, in descending order: car travel, cycling, and walking, with ratios of 62.7%, 28.7%, and 7.7%, respectively, indicating a significant gap between the service area ratios of public fitness venues among the three travel modes. At the same time, there is no apparent relationship between the accessibility of public fitness venues and the demographic and economic elements in the main urban area of Jinan, indicating that the construction of public fitness venues does not fully consider the influence of population or economy on accessibility. Moreover, our findings echo the views of other scholars on urban public fitness venue accessibility studies; therefore, it can be concluded that uneven spatial distribution of public fitness venues and fairness to be improved are common problems. At the same time, the present study's focus on the influence of public fitness venue accessibility from the perspective of residents' travel choice of walking, cycling, or car travel precisely complements the research gaps in this field; furthermore, the analysis based on objective data including road networks ensures that the results are objective and provide a scientific reference value.

**6. Suggestions**

Based on the spatial distribution characteristics of public fitness venues in the main urban area of Jinan, it is recommended to increase the construction of public fitness venues in the surrounding areas of the main urban area, which are weak in terms of both quantity and density in this regard. Therefore, the construction of public fitness venues must first increase in number, and at the same time spread to the surrounding areas in a targeted manner to meet the fitness needs of urban residents in non-central areas. Secondly, based on the accessibility of public fitness, it is recommended that planners consider both population and the distribution of various age groups in the region, and focus on increasing the construction of public fitness venues in areas with poor per capita public fitness venues. Finally, it is suggested that the government should analyze the travel modes and habits of urban residents based on the conclusions of this study, so as to carry out scientific site selection and planning of public fitness venues.

**Author Contributions:** Conceptualization, Y.J. and H.Z.; methodology, D.S.; software, C.W.; validation, Y.J., Z.L. and W.S.; formal analysis, D.S.; investigation, Z.S.; resources, Y.J.; data curation, Z.L.; writing—original draft preparation, Y.L.; writing—review and editing, H.Z.; visualization, C.W.; supervision, Z.S.; project administration, X.W.; funding acquisition, Y.J. All authors have read and agreed to the published version of the manuscript.

**Funding:** This study was supported by the following funds. (1) Basic Scientific Research Project of Education Department of Liaoning Province: Research on the overall development of "Competition + Industry" Goal of Liaoning Province Three Big Globe Project (grant numbers: LJKMR20221375) (2) Economic and Social Development Research Project of Liaoning Province: Research on Social Support System of Youth Sports and Fitness Behavior from the Perspective of Family Endowment (grant numbers: 2023LSLYBKT-082).

**Institutional Review Board Statement:** Not applicable.

**Informed Consent Statement:** Not applicable.

**Data Availability Statement:** Not applicable.

**Acknowledgments:** The authors would like to acknowledge all colleagues and friends who have voluntarily reviewed the translation of the survey and the manuscript of this study.

**Conflicts of Interest:** The authors declare no conflict of interest.

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
