# Peer review of "Spatial Distribution Characteristics of Public Fitness Venues: An Urban Accessibility Perspective"

_sustainability, doi:10.3390/su15010601_

Round 1
Reviewer 1 Report
The paper analyzes the spatial distribution characteristics and accessibility of public fitness venues in Jinan city based on three travel modes of walking, cycling, and car. This method is useful for understanding and assessing the amounts, distribution, and accessibility of public fitness venues. However, the following questions are required to justify:
1. What's the theoretical and methodological novelty of this paper compared to the authors’ recently published paper "Spatial Distribution Characteristics of Public Fitness Venues in the Main Urban Area of Dalian from the Perspective of Urban Accessibility" (https://www.mdpi.com/2071-1050/14/19/12728)? It shows a high similarity between these two papers, no matter the research background, literature review, or workflow.
2. What are the specific research questions of this paper? And what specific research gaps do the authors want to bridge?
3. How do identify PUBLIC FITNESS VENUES? What are the supporting reasons why authors selected venues larger than 10 km2 as the study samples?
4. What’s the representativeness of Jinan city as a case study?
Besides, the following adjustments are required:
Abstracts: Abstracts should be revised to be concise and factual - briefly stating the purpose of the research, the principal results, major conclusions, and suggestions, rather than tediously long data.
Introduction:
It’s required to justify the core research questions and the brief introduction of respective methods. In addition, the literature review should state the research gaps at the end of each session.
Results:
In each session of the data analysis, relevant explanations are needed, rather than only stating the results of the data analysis.
Suggestions:
Based on the results of spatial distribution and accessibility characteristics of public fitness venues, it’s better to give some urban planning or decision-making suggestions at the end.
language errors:
1. The public fitness venues selected are larger than 10 km2 or 10 m2? The descriptions are different in the abstract and the main contents.
2. Subtitles of 3.1.1, 3.2.2, 3.3.3 are the same
3. Double-check for other grammar and writing errors
Reviewer 2 Report
This paper showed the spatial characters of Public Fitness Venues based on the perspective of accessibility. There were interesting results about the relationship. The scientific problems of this paper were well raised and responded. However, on the whole, the research method is relatively simple and needs to be refined.
1. The preface is lengthy and needs to be simplified, mentioning the specific research progress and results of the current research on accessibility.
2. The research method is relatively simple, and its innovation needs to be improved. It is suggested that population data on accessibility coverage should be added, equity analysis should be added, and Public Fitness Venues classification should be carried out by area or by level to obtain more guidance results.
3. As for the current research on accessibility, it is necessary to analyze the distribution characteristics of walking roads, cycling roads and vehicular roads in the whole city. Based on this, relevant conclusions can be drawn by combining walking, cycling and vehicular roads.
4. In the discussion section, it is suggested to discuss the main results obtained, and the relevant conclusions should be compared with the results of previous studies, so as to discuss the significance of the results of this paper.
Some details need to be revised
1. English language and style are fine/minor spell check required.
2. The resolution figures, such as Figure 1 and Figure 3, should be improved for the convenience of readers.
3. The data sources in Table 1 need to indicate the website or literature sources.
4. The data presentation in Figure 4 raises some doubts. Does it mean that the Public Fitness Venues in a certain block are of the same size? These figures do not accord with the objective facts. If it's about calculating the average Public Fitness Venues in a certain area, I don't think this analysis has strong scientific significance.
5. The title is not rigorous. The title below 3.3 is 3.1.1? The title of 3.1.1 is the same as that of 3.3.2 and 3.3.3, but the content is different. It is the accessible area of walking, cycling and car shops respectively, which needs to be sorted out again.
Reviewer 3 Report
The research topic is interesting and of value to discuss in deep, which will imply future urban planning. However, the research method is not scientific enough and the discussions need to be modified. The conclusions are very important and mandatory.
1. Abstract:
Highlight the problems. Conclude your results and analysis, not simply described. It will be good if you can write your contributions and implications.
2: Introduction:
Structures could be more logic. Please modify it.
3. Methods:
Current methods are not scientific enough. Would be good if you can consider populations and economic characteristics by using ArcGIS spatial analysis.
4. Results:
Figures are clear. Would you please add the complete legends, not only the classifications of colours?
5. Discussions:
"Authors should discuss the results and how they can be interpreted from the perspective of previous studies and of the working hypotheses. The findings and their implications should be discussed in the broadest context possible. Future research directions may also be highlighted."
Is it the requirement in the template?
You only described the table but didn't analyse the results. Please analyse quantitively and scientifically.
Your study was based on three modes of transportation, then why you only considered car travel to analyse? I think it will be good if you can analyse these three modes and compare the results.
Line 402-416: It's the related work, not your discussions.
Please discuss in deep.
6. Conclusions:
"This section is not mandatory but can be added to the manuscript if the discussion is unusually long or complex."
Conclusion is definitely mandatory. It is not a repeat of your discussion. You should conclude your methods, results, analysis and discussions. It is also necessary to highlight your contributions and implications.
Please rewrite the conclusions.
Round 2
Reviewer 1 Report
the manuscript is ready to publish in its present form in my view.